# Global profiling of protein–DNA and protein–nucleosome binding affinities using quantitative mass spectrometry

Matthew M. Makowski[1,2], Cathrin Gräwe[1,2], Benjamin M. Foster[3,4,5], Nhuong V. Nguyen [4,5], Till Bartke[3,4,5] & Michiel Vermeulen [1,2]

Interaction proteomics studies have provided fundamental insights into multimeric biomolecular assemblies and cell-scale molecular networks. Significant recent developments in mass spectrometry-based interaction proteomics have been fueled by rapid advances in label-free, isotopic, and isobaric quantitation workflows. Here, we report a quantitative protein–DNA and protein–nucleosome binding assay that uses affinity purifications from nuclear extracts coupled with isobaric chemical labeling and mass spectrometry to quantify apparent binding affinities proteome-wide. We use this assay with a variety of DNA and nucleosome baits to quantify apparent binding affinities of monomeric and multimeric transcription factors and chromatin remodeling complexes.

[1] Department of Molecular Biology, Radboud Institute for Molecular Life Sciences, Radboud University, Nijmegen 6500 HB, The Netherlands. [2] Oncode Institute, Radboud University, Nijmegen 6500 HB, The Netherlands. [3] Institute of Functional Epigenetics, Helmholtz Zentrum München, 85764 Neuherberg, Germany. [4] MRC London Institute of Medical Sciences (LMS), London W12 0NN, UK. [5] Institute of Clinical Sciences (ICS), Faculty of Medicine, Imperial College London, London W12 0NN, UK. These authors contributed equally: Cathrin Gräwe, Benjamin M. Foster, Nhuong V. Nguyen. Correspondence and requests for materials should be addressed to T.B. (email: till.bartke@helmholtz-muenchen.de) or to M.V. (email: michiel.vermeulen@science.ru.nl)

I nteraction proteomics, via mass spectrometry, has contributed invaluably to the identification of physical associations between biological molecules and the translation thereof into cellular protein networks[1–3]. Interaction proteomics methodologies are generally semi-quantitative and use label-free or chemical labeling-based relative quantification to call interactions as outliers from a background of non-specific identifications[4,5]. Thus, interactions are regularly reported in a binary on/off manner, though semi-quantitative stoichiometric information is beginning to add a quantitative dimension to interaction studies. Nevertheless, a complete characterization of a functioning cell requires knowledge not only of specificity of biomolecular interactions (i.e., does an interaction occur, or not) but also of affinity (i.e., how strong, in absolute terms, is some given interaction)[6]. Typical affinity quantitation methods, such as isothermal titration calorimetry, surface plasmon resonance, fluorescence polarization, fluorescence resonance energy transfer, or electrophoretic mobility shift assay (EMSA), have been performed on a single interaction, case-by-case basis, and require laborious expression and purification of recombinant proteins. Importantly, chemoproteomic approaches studying protein–small molecule interactions established the possibility of designing absolutely quantitative binding assays using semi-quantitative isobaric labeling and mass spectrometry[7,8]. Similarly, thermal proteome profiling is an innovative mass spectrometry approach that uses thermal stability shifts upon small-molecule binding to estimate apparent dissociation constants proteome-wide, again with a semi-quantitative isobaric labeling strategy[9,10]. Other protein-centric studies have focused on measuring the DNA-binding landscape of a single, or a few, transcription factors[11,12]. However, the inverse problem of interrogating the quantitative protein binding landscape of DNA sequences of interest has received considerably less attention and is still dominated by semi-quantitative workflows[13,14]. Here, we present a method for determining, proteome-wide, tens to hundreds of apparent dissociation constants ($K_d^{APP}$) of nuclear proteins for DNA and nucleosome ligands simultaneously using affinity purification from nuclear lysates and isobaric 10-plex tandem mass tag (TMT) labeling coupled with mass spectrometry.

## Results

### Benchmarking $K_d^{APP}$ measurements with the SP/KLF motif.
We first affinity purified nuclear proteins from isolated nuclear lysates using a series of ten pulldowns with different concentrations of oligonucleotide baits coupled to streptavidin–sepharose beads. Proteins binding at each titration point were digested to tryptic peptides and isobarically labeled with the 10-plex TMT system[15]. Labeled peptides were combined and measured in a single synchronous precursor selection-MS3 (SPS-MS3) mass spectrometry run[16,17] (Fig. 1a). Critically, the highest titration point (labeled with TMT131) represented a pulldown at micromolar concentration. We assumed that proteins with nanomolar range apparent dissociation constants ($K_d^{APP} \sim 1–500$ nM) would exhibit saturated binding at this concentration. Thus, for each DNA concentration we calculated the bound fraction for each individual protein compared to the TMT131 reporter ion signal (Fig. 1b and Supplementary Fig. 1). $K_d^{APP}$ values were determined independently for each protein by fitting the parameters of a Hill-like curve using the known DNA concentrations and the observed fraction bound (Fig. 1a). We filtered out background or non-specific proteins based on the quality of fit of the Hill curve, under the assumption that background proteins would show randomly distributed ratios near 1:1 for all titration points. As such, only proteins fitting a Hill-like curve with an $r^2$ value >0.95 were kept for downstream analysis. As each bait profiled required a set of ten pulldowns conducted in triplicate or duplicate for fitting, we utilized a filter plate system to increase throughput[18]. We note that this system is amenable to automation in future high-throughput studies.

As a benchmark case, we quantitatively profiled the nuclear protein binding landscape of the well-characterized SP/KLF consensus GC-box[18]. Oligo depletion was essentially complete after bead immobilization (Supplementary Fig. 2A). Binding of canonical SP/KLF factors was strongly specific for the designed SP/KLF motif oligonucleotide, with essentially no background binding observed for a mutated SP/KLF motif (Supplementary Fig. 2B). Using an $r^2$ value of 0.95 as a filtering criterion, we observed low coefficients of variation for fitted $K_d^{APP}$ values (Fig. 1c). We estimated the $K_d^{APP}$ value for canonical SP/KLF binding factor SP1 as ~38 nM and confirmed this result in lysates with gel-based assays (Fig. 1d and Supplementary Fig. 2C). Furthermore, we estimated $K_d^{APP}$ values for a number of other SP/KLF family factors including SP3 and KLF4. Interestingly, purified recombinant SP3 exhibited a substantially lower $K_d$ in gel-shift assays indicating a higher affinity compared to mass spectrometry-based measurements from nuclear lysates (Supplementary Fig. 2D). However, this shift was abrogated when recombinant SP3 was spiked into nuclear lysates, where recombinant and endogenous SP3 showed comparable binding curves. Similarly, we observed a lower $K_d$ value for the SP/KLF motif and purified recombinant KLF4-ZF domain using fluorescence polarization and fluorescence de-quenching assays compared to those measured for endogenous KLF4 in lysates by mass spectrometry and gel-based assays (Supplementary Fig. 3)[19,20]. This clearly suggests competitive inhibition between proteins in the nuclear environment as has been reported for SP1-KLF4 and SP1-KLF16 among other SP/KLF factors[21]. As such, this complex interplay between DNA binding proteins indicates $K_d^{APP}$ measurements will be useful information for uncovering interactions within transcriptional networks as they exist in the in vivo nuclear environment.

To further characterize the specificity and sensitivity of this assay, we conducted a series of competition experiments with free (unbiotinylated) wild-type and mutated SP/KLF oligonucleotides. In agreement with western blot analysis (Supplementary Fig. 2B), known sequence-specific SP/KLF transcription factors showed no measurable binding to the mutated SP/KLF oligonucleotide (Supplementary Fig. 4A). In competition experiments, these sequence-specific factors were similarly not competed away from immobilized wild-type oligonucleotides by free mutated oligonucleotides (Supplementary Fig. 4A). Yet, importantly, $K_d^{APP}$ values for sequence-specific proteins estimated by applying the Cheng–Prusoff correction to $IC_{50}$ values from competition experiments (Supplementary Fig. 4B) were highly correlated to $K_d^{APP}$ values estimated with immobilized oligonucleotides at a near 1:1 ratio (Supplementary Fig. 4C). In contrast, non-sequence-specific proteins bound to immobilized wild-type or mutated SP/KLF oligos with high correlation (Supplementary Fig. 4A). Non-sequence-specific proteins were also competed from immobilized wild-type oligonucleotides by either wild-type of mutated free oligonucleotide. However, non-sequence-specific proteins displayed lower $K_d^{APP}$ values on average to either free oligonucleotide compared to immobilized oligonucleotides in competition experiments (Supplementary Fig. 4C). We attribute this to free oligonucleotides having two available blunt ends acting as substrates for some non-sequence-specific proteins, while immobilized oligonucleotides only have one sterically free blunt end. This is an important consideration for future studies comparing free vs. immobilized oligonucleotides and sequence-specific vs. non-specific DNA-binding factors.

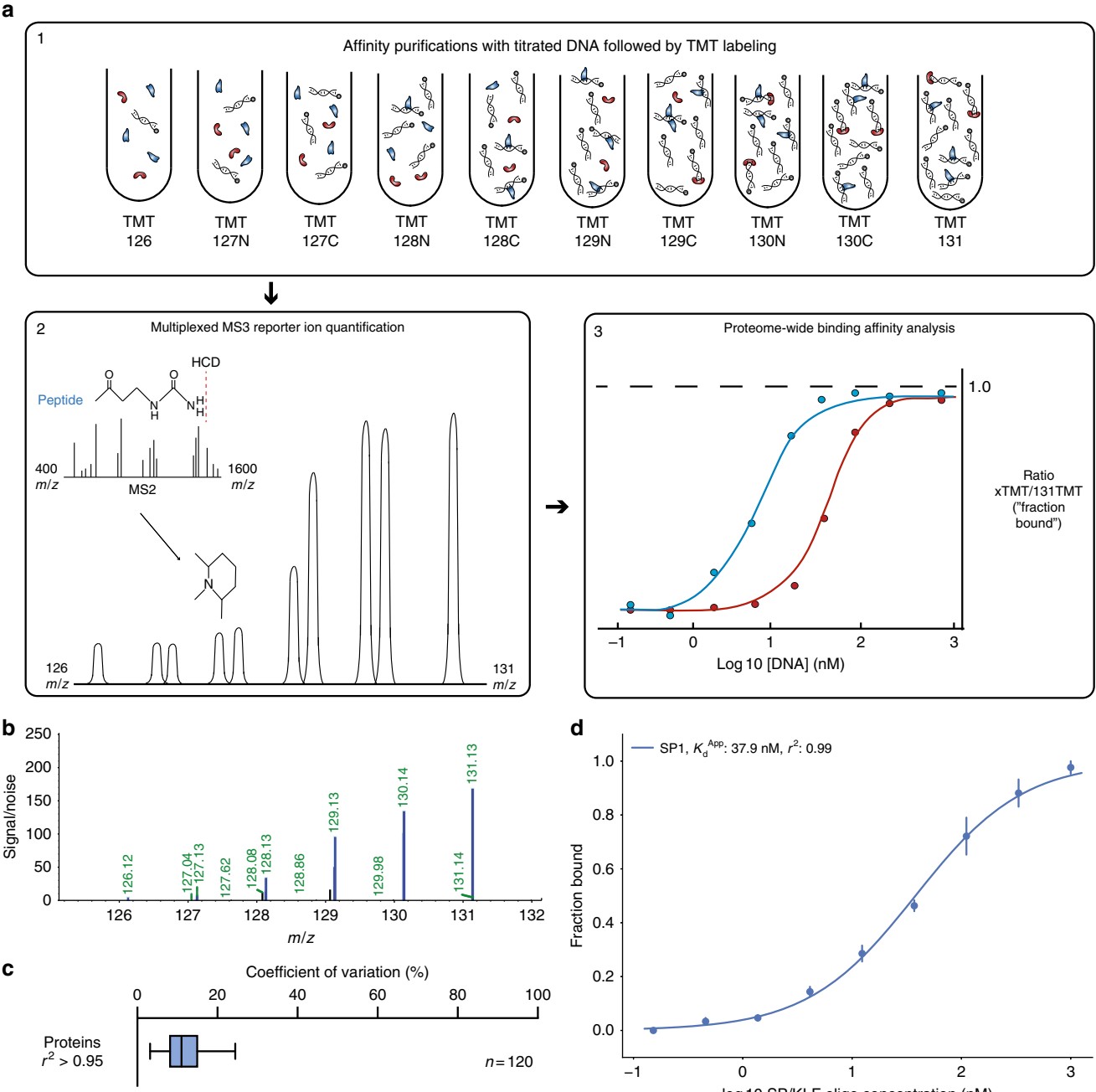

**Fig. 1** Benchmarking protein–DNA $K_d^{App}$ measurements with the SP/KLF consensus motif. **a** A titration series of a known concentration of bait is used for affinity purification of proteins from nuclear lysates. Bound proteins are digested with trypsin, isobarically labeled with TMT reagent, and analyzed by mass spectrometry. Quantification of binding interactions yields a Hill-like curve, as described in the Methods, which can be used to calculate the $K_d^{App}$. **b** SPS-MS3 TMT reporter ion spectrum of an example SP1 peptide. Only the low $m/z$ range of the MS3 spectrum, where the TMT reporter ions are observed, is displayed for clarity. Plotted on the y-axis are signal-to-noise values measured in the orbitrap at 60,000 resolution. **c** Boxplot analysis of all coefficients of variation for fitted $K_d^{App}$ values identified using the SP/KLF consensus motif with $r^2$ values >0.95. The box represents the quartiles of the data, while the whiskers represent the range of 1.5 IQRs. The center line is the median of the distribution. **d** Hill-like curve identified for SP1 binding to the consensus SP/KLF GC-box motif. Binding curves were generated by fitting the parameters of the Hill equation including $K_d^{App}$. Each data point is the mean of three experiments ($n=3$), and the error bars represent the standard error of the mean

The $K_d^{App}$ values we measured did not significantly correlate with absolute protein abundance, demonstrating that the $K_d^{App}$ values we measured were not biased by protein abundance (Supplementary Fig. 4D and Supplementary Data 2)[22,23]. An additional analysis taking possible phospho-post-translational modifications into account also showed practically no difference with our original result, suggesting such phospho-modifications, even if stoichiometric in vivo, may not be easily identified without specific enrichment methods (Supplementary Fig. 4E)[24]. Finally, we calculated JASPAR transcription factor binding profile motif scores for sequence-specific proteins with JARPAR motifs and correlated them with the $K_d^{App}$ values we measured (Supplementary Fig. 5). We observed significant correlation, demonstrating that our assay produced results consistent with

motif-based binding landscape models. Summarily, on these bases we concluded that our assay generated reliable $K_d^{App}$ measurements for protein–DNA interactions in this SP/KLF benchmark case.

**Chromatin remodelers quantitatively prefer G-quadruplexes.** Next, we conducted a larger assay of eight different double-stranded DNA (dsDNA) or single-stranded DNA (ssDNA) sequences (Fig. 2a and Supplementary Table 1) representing canonical, well-characterized biological motifs (AP-1, CTCF, E-box, NF-Y, TATA, TEAD, the human ssDNA telomere repeat, and the c-Myc promoter Pu27 G4-quadruplex forming ssDNA sequence [mycG4][25]). We observed numerous homo-multimeric and hetero-multimeric transcription factors and DNA-binding proteins binding to their canonical motif in readily distinguishable clusters, including JUNB/JUND, MAX, POT1, and TEAD/YAP (Fig. 2a–e). More specifically, we were surprised and intrigued to see subunits of many chromatin-modifying complexes binding to the mycG4 sequence in G4-permissive NaCl-based binding buffer, including SWItch/Sucrose Non-

Fermentable (SWI/SNF) and Imitation SWI (ISWI) subunits (Fig. 2f, g). This immediately suggested the possibility that some chromatin remodeling and modifying enzymes specifically recognize DNA G4 structures. To further characterize these interactions, we performed an additional set of assays using either 150 mM LiCl buffer, which destabilizes G-quadruplex structures, or 150 mM NaCl binding buffer supplemented with 20 μM PhenDC3, which stabilizes G4 structures (Fig. 2a). Both LiCl and G-quadruplex stabilizing small molecules were shown previously to inhibit G4-RNA binding of polycomb repressive complex 2 (PRC2)[26], which binds RNA G-quadruplexes through interfaces on EZH2 and EED subunits and an RNA recognition motif in the SUZ12 subunit[27,28]. Here, we observed that $K_d^{App}$ values measured for the mycG4 sequence in LiCl and PhenDC3 binding conditions were indeed positively correlated (Supplementary Fig. 6A). LiCl or PhenDC3 treatment quantitatively and significantly increased $K_d^{App}$ values for both SWI/SNF and ISWI subunits and for PRC2 and nucleosome remodeling and deacetylase (NuRD) subunits (Fig. 2f, g and Supplementary Fig. 6B, C). Among these, PRC2 and NuRD subunits bound with ~100 nM $K_d^{App}$s to the mycG4 sequence, similar to the recently

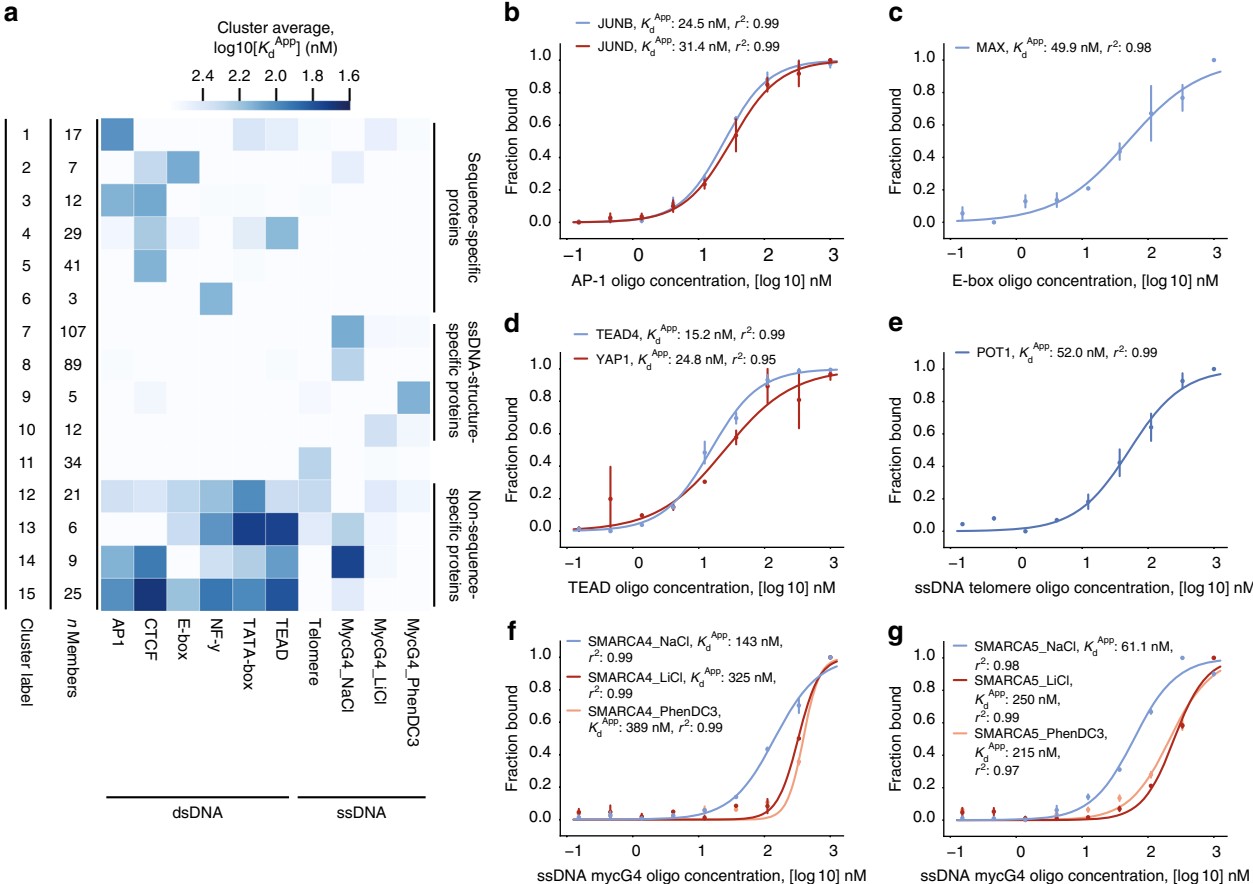

**Fig. 2** A motif survey identifies SWI/SNF and ISWI factors binding to G4-quadruplex structures. **a** Heatmap analysis $K_d^{App}$ binding profiles for all dsDNA and ssDNA sequences and experiments. Proteins were clustered (15 clusters) using hierarchical agglomerative clustering. The heatmap is colored by the average log10[$K_d^{App}$] value of the cluster per bait. Cluster labels and number of proteins per cluster ($n$) are listed in columns to the left of the heatmap. **b–g** $K_d^{App}$ binding curves for canonical and unreported binding proteins for some example dsDNA and ssDNA motifs. **b** $K_d^{App}$ binding curve for AP-1 dsDNA motif and dimeric binding factors JUNB (Cluster 15) and JUND (Cluster 3). **c** $K_d^{App}$ binding curve for E-box dsDNA motif and dimeric binding factor MAX (Cluster 2). **d** $K_d^{App}$ binding curve for TEAD dsDNA motif and dimeric binding factors TEAD4 (Cluster 4) and YAP1 (Cluster 4). **e** $K_d^{App}$ binding curve for the telomere ssDNA motif (four repeats) and binding factor POT1 (Cluster 11). **f** $K_d^{App}$ binding curve for the mycG4 ssDNA motif in NaCl (G4-permissive), LiCl (G4-nonpermissive), and PhenDC3 (G4-ligand) binding conditions and SWI/SNF binding factor SMARCA4 (Cluster 7). **g** $K_d^{App}$ binding curve for the mycG4 ssDNA motif in NaCl, LiCl, and PhenDC3 binding conditions and ISWI binding factor SMARCA5 (Cluster 7). For **b–g**, binding curves were generated by fitting the parameters of the Hill equation including $K_d^{App}$. Each data point is the mean of two experiments ($n = 2$), and the error bars represent the standard error of the mean

reported binding affinity of PRC2 to RNA G-quadruplexes[26,27]. Further supporting this finding, we detected enrichment of induced G-quadruplex structures in both SWI/SNF, PRC2, and NuRD subunit binding sites in a variety of ENCODE ChIP-sequencing (ChIP-seq) datasets, consistent with our in vitro binding data (Supplementary Fig. 6D)[29–31]. We also verified this finding by western blot (Supplementary Fig. 7A–C). Finally, we were interested in identifying potential direct G4 DNA-binding subunits from these chromatin remodeling and modifying complexes. We adapted a formaldehyde cross-linking approach to identify individual peptides immediately proximal to the mycG4 DNA bait (Supplementary Fig. 8A)[32]. Identified peptides from a known DNA-binding complex were proximal to the DNA-binding channel, indicating the reliability of the approach (Supplementary Fig. 8B, PDB: 1JEY (https://doi.org/10.2210/pdb1JEY/pdb))[33]. We identified a peptide from the N-terminal winged helix-like domain (WHD) of SMARCB1 as the most significantly enriched for direct G4 binding of SWI/SNF, ISWI, PRC2, and NuRD subunits (Supplementary Fig. 8C–E). Affinity purification experiments with the recombinant SMARCB1-WHD verified that the SMARCB1-WHD specifically recognizes the mycG4 bait compared to a variety of control baits while the double plant homeodomain (PHD) finger of PHF10, another SWI/SNF accessory subunit, does not (Supplementary Fig. 8F)[34]. Overall, these in vitro analyses provide biochemical support for the hypothesis that some chromatin remodelers and modifiers can bind G4 DNA sequences in a G-quadruplex preferential manner.

### $K_d^{App}$ estimation using nucleosome substrates.

Finally, to show that this binding assay is compatible not only with nucleic acids but also with nucleoprotein complexes, we performed a set of experiments using mono-nucleosomes, di-nucleosomes, and modified di-nucleosomes (Fig. 3a and Supplementary Fig. 9A–D). We identified a number of protein–di-nucleosome interactions that were either specific to or modulated by either H3K4me3 or H3K9AcK14Ac (Fig. 3b, d, e). Across all baits, coefficients of variation were reasonably low indicating good data quality (Fig. 3c). We observed SWI/SNF and ISWI binding with significantly lower $K_d^{App}$ to H3K9AcK14Ac-containing di-nucleosomes compared to unmodified di-nucleosomes (Supplementary Fig. 10A, D). Indeed, many accessory SWI/SNF subunits were uniquely identified with the H3K9AcK14Ac di-nucleosome bait. We identified one SWI/SNF subunit, PHF10, which harbors a C-terminal double PHD finger (DPF) and may preferentially interact with H3K9AcK14Ac compared to unmodified H3 or H3K4me3 as has been reported for another DPF domain[35]. PHF10 shows one of the lowest $K_d^{App}$ values of all identified SWI/SNF subunits (~35 nM, Fig. 3e) for H3K9AcK14Ac di-nucleosomes and was similarly identified exclusively with the H3K9AcK14Ac di-nucleosome. To further investigate the H3K9AcK14Ac specificity of PHF10, we used bacterial lysates expressing the recombinant glutathione S-transferase (GST)-tagged DPF domain of PHF10 in histone H3 peptide pull-down experiments (Supplementary Fig. 10B). This experiment clearly confirmed that recombinant PHF10-DPF binding to H3 is strongly agonized by H3K9AcK14Ac in contrast to the SMARCB1-WHD, as has been reported for the bromodomain of Swi2/Snf2 and the DPF domains of SWI/SNF accessory subunits DPF2 and DPF3 (Supplementary Fig. 10C)[36–38]. Thus, PHF10 represents an additional high affinity H3 acetylation reader in the human SWI/SNF complex. Intriguingly, PHF10-DPF seems to be repelled by H3K4me3, and our data suggests this might confer a lower affinity fine-tuning on PHF10-SWI/SNF nucleosome binding. However, even in the absence of the acetylation specificity conferred by SWI/SNF acetylation reader domains, we

observed that catalytic SWI/SNF subunits SMARCA2 and SMARCA4 still engage in relatively high affinity interactions with H3K4me3 modified and unmodified di-nucleosomes (Supplementary Fig. 10A, D). Similarly, we observed relatively high affinity interactions between modified or unmodified di-nucleosomes and ISWI catalytic subunit SMARC5 and ISWI accessory subunits including BPTF, C17orf49 (Fig. 3e), BAZ1A, and BAZ1B. The binding patterns we observe between different di-nucleosome baits are complex and highlight an important point and a unique benefit of measuring apparent affinities from complex lysates: we measure the average binding profile over what is likely a pool of heterogeneous multimeric protein complexes. For example, we observe catalytic subunit SMARCA5 binding with highest affinity to H3K9AcK14Ac di-nucleosomes compared to unmodified or H3K4me3 di-nucleosomes (Supplementary Fig. 10D). Yet, accessory subunits BPTF, C17orf49, BAZ1A, and BAZ1B exhibit binding curves that are both similar to and distinct from the binding curve of SMARCA5 depending on the accessory subunit and di-nucleosome pair in question, suggesting these subunits regulate differential H3 modification-specific binding (Supplementary Fig. 10D). Moreover, these ISWI accessory subunits, along with SMARCA5, form at least three unique and independent protein complexes (the NuRF, ACF, and WICH complexes)[39]. More generally, deconvoluting the individual contributions of different subcomplexes with recombinant systems is indisputably critical (e.g., as with SMARCB1 and its G-quadruplex interaction and PHF10-H3K9AcK14Ac binding in SWI/SNF). In the future, a data-driven strategy for assessing the contributions of shared subunits within independent subcomplexes might involve a combinatorial approach utilizing many DNA or modified nucleosome baits as was demonstrated, for example, with the stoichiometry of different SET1/MLL subcomplexes[22]. Additionally, we argue that the ability to measure binding affinities for multimeric complexes in the complex environment of the nucleus offers an important holistic, system-wide view.

## Discussion

By measuring protein–DNA and protein–nucleosome $K_d^{App}$ values via mass spectrometry, we provide another avenue for extending protein–nucleic acid interaction proteomics beyond comparative, semi-quantitative workflows. From a systems biology perspective, absolute binding affinities within the nuclear environment create a quantitative link between transcription factor expression and target gene regulation via the TF–DNA interaction. However, our data imply a situation where DNA sequences, structures, and chromatinized DNA integrate signals from a variety of binding partners at a spectrum of biologically relevant affinities. Indeed, we observe that both DNA structures, including G-quadruplexes, and (modified) nucleosomes engage in high affinity-binding interactions with various independent chromatin remodeling and modifying complexes. Of note, we observe SWI/SNF as a complex that recognizes both DNA structure and histone modification state, and we identify SMARCB1 and PHF10, respectively, as specific readers that contribute to these high affinity substrate recognitions. While targeted biochemical analysis reveals direct mediators of specific substrate interactions within a complex, combining these analyses with affinity measurements in the context of multimeric assemblies in complex lysates takes into consideration the possibility of larger molecular regulatory networks. This work demonstrates an assay for revealing not only these regulatory interactions but also their absolute affinities and thereby profiling the proteome-wide quantitative binding landscape of nuclear proteins for DNA oligonucleotides and nucleosome complexes.

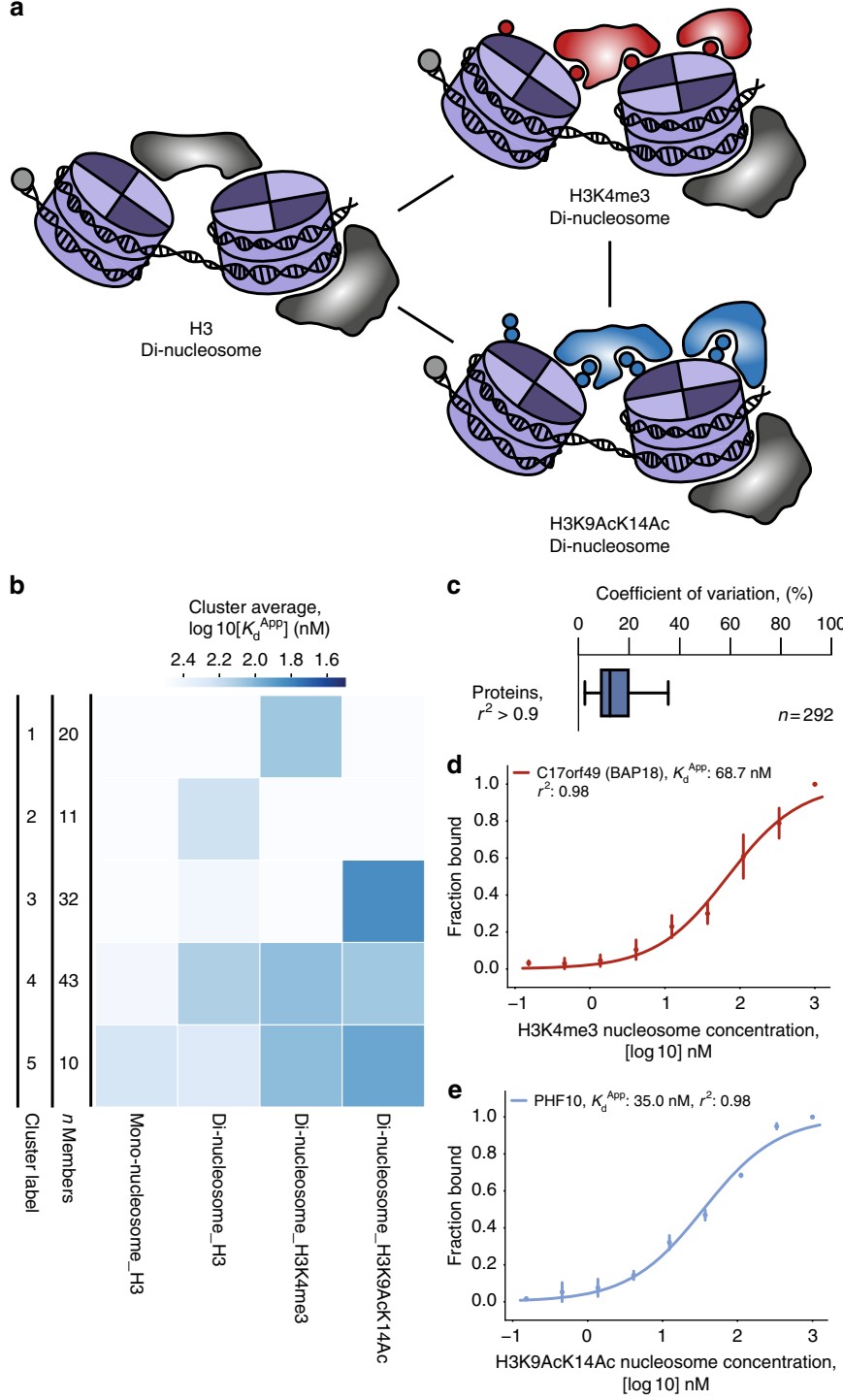

**Fig. 3** Quantitative analysis of modified di-nucleosome interactions. **a** Schematic representation of $K_d^{App}$ di-nucleosome and modified di-nucleosome study design. **b** Heatmap analysis of $K_d^{App}$ binding profiles for all nucleosome, di-nucleosome, and modified di-nucleosome experiments. Proteins were clustered (five clusters) using hierarchical agglomerative clustering. The heatmap is colored by the average $\log 10[K_d^{App}]$ value of the cluster per bait. Cluster labels and number of proteins per cluster ($n$) are listed in columns to the left of the heatmap. **c** Boxplot analysis of all coefficients of variation for fitted $K_d^{App}$ values identified in nucleosome experiments with $r^2$ values >0.90. The box represents the quartiles of the data, while the whiskers represent the range of 1.5 IQRs. The center line is the median of the distribution. **d**, **e** Example $K_d^{App}$ binding curves for binding proteins of H3K4me3 and H3K9AcK14Ac di-nucleosomes. **d** $K_d^{App}$ binding curve for H3K4me3 modified di-nucleosomes and binding factor C17orf49 (Cluster 1). **e** $K_d^{App}$ binding curve for H3K9AcK14Ac modified di-nucleosomes and binding factor PHF10 (Cluster 3). Binding curves were generated by fitting the parameters of the Hill equation including $K_d^{App}$. Each data point is the mean of three experiments ($n = 3$), and the error bars represent the standard error of the mean

## Methods

**Cell culture and nuclear lysate preparation.** Wild-type HeLa Kyoto cells (received from Anthony Hyman and Ina Poser, Human HeLa BAC database, Dresden, Germany) were cultured in Dulbecco's modified Eagle's medium supplemented with 10% fetal bovine serum and 100 U/mL penicillin and streptomycin. Cells were periodically tested for mycoplasma contamination. Nuclear lysates were isolated as described previously[14]. Briefly, cells were lysed by swelling and mechanical force in buffer A (10 mM HEPES (pH 7.9), 1.5 mM MgCl2, 10 mM KCl, and 0.15% NP40). Then, nuclei were collected by centrifugation and chemically lysed in buffer C (420 mM NaCl2, 20 mM HEPES (pH 7.9), 20% (v/v) glycerol, 2 mM MgCl2, 0.2 mM EDTA, 0.1% NP40, EDTA-free complete protease inhibitors (CPIs, Roche), and 0.5 mM dithiothreitol (DTT)). Protein concentrations were assessed by Bradford assay.

**Affinity purification and sample preparation.** Oligonucleotides for affinity purification were ordered as custom-synthesized oligos from Integrated DNA Technologies (IDT; Supplementary Table 1). DNA and nucleosome affinity purifications were performed using a filter plate-based workflow described first by Hubner et al[18]. The essential protocol is described below.

First, all dsDNA oligos were annealed by heating to 95 °C for 10 min before cooling to room temperature. Each oligo was then diluted to a working stock of 3 µM, which represented the highest concentration reference titration point in this study. A series of nine three-fold dilutions was then prepared, resulting in a titration series of 10 oligo concentrations ranging from 0.15 nM to 3 µM. We prepared 200 µL of oligo per titration point per replicate in this fashion. Dilutions were performed in DNA binding buffer (DBB: 1 M NaCl, 10 mM Tris, pH 8, 1 mM EDTA, 0.05% NP-40).

Filter plates were first prepared with 50 µL of ethanol per well (96-well filter plate, 1.2 µM pore, Millipore/Merck MSBVS1210). Wells were then washed twice with DBB. Twenty microliters of streptavidin–sepharose bead slurry was added to each well (GE, 10 µL beads, 3 nmol binding capacity). Wells were washed twice with DBB. One hundred and fifty microliters of the corresponding oligo titration point was added to each well, and oligos were immobilized to the streptavidin-conjugated beads over a 1 h incubation at 4 °C while shaking on a tabletop microplate shaker. Samples were then washed once with DBB and twice with protein binding buffer (PBB: 150 mM NaCl, 50 mM Tris, pH 8, 0.25% NP-40, 1 mM Tris(2-carboxyethyl)phosphine (TCEP), and CPIs). One hundred micrograms of HeLa nuclear lysate was then diluted to 150 µL final volume in PBB and added to each well. Samples were incubated for 2 h at 4 °C while shaking on a microplate shaker, and then washed six times with washing buffer (150 mM NaCl, 100 mM triethylammonium bicarbonate, TEAB).

For competition experiments using SP/KLF wild-type and mutated oligonucleotides, beads were pre-washed as described and pre-incubated in 96-well plate format with 200 nM biotinylated SP/KLF wild-type oligonucleotide diluted in DBB for 1 h at 4 °C while shaking on a tabletop microplate shaker. Free (unbiotinylated) wild-type or mutated SP/KLF oligonucleotide were prepared at three-fold diluted concentrations in PBB from 0.3 nM to 6 µM as described above. One hundred micrograms of HeLa nuclear lysate was prepared in PBB and oligonucleotides were mixed with HeLa nuclear lysates at a ratio of 1:1 in a final volume of 150 µL, with free oligonucleotides at a final concentration of 0.15 nM to 3 µM. Samples were then washed once with DBB and twice with PBB. HeLa lysates plus free oligonucleotides were added to the beads containing 200 nM immobilized SP/KLF oligonucleotide. Proteins were incubated for 2 h at 4 °C while shaking on a microplate shaker, and then washed six times with washing buffer. Sample preparation for mass spectrometry was continued as described below.

For the LiCl mycG4 experiment, the oligos were heated for 10 min at 95 °C and afterwards snap-cooled on ice for 3–5 min before they were immobilized on the beads. The NaCl in the PBB was replaced with 150 mM LiCl for the oligo titration, protein incubation, and washing steps (PBB composition: 150 mM LiCl, 50 mM Tris, pH 8, 0.25% NP-40, 1 mM TCEP, and CPIs). For the PhenDC3 mycG4 experiment, 20 µM PhenDC3, final concentration, was added to the PBB for oligo titration, protein incubation, and washing steps (PBB composition: 150 mM NaCl, 50 mM Tris, pH 8, 0.25% NP-40, 1 mM TCEP, CPIs, and 20 µM PhenDC3). For the nucleosome experiments, nucleosome titration and immobilization was performed in PBB instead of DBB. All other experimental conditions were the same as described above.

Sample preparation for mass spectrometry was performed by first adding 50 µL of elution buffer (20% methanol, 80 mM TEAB, 10 mM TCEP). Proteins were incubated for 30 min at room temperature, alkylated with 50 mM iodoacetamide, and digested with 0.25 µg trypsin overnight while shaking on a tabletop shaker at room temperature.

For formaldehye cross-linking experiments using the mycG4 ssDNA oligonucleotide, 500 µg of HeLa nuclear lysate was prepared in a volume of 600 µL borate-buffered saline (50 mM boric acid, pH 8.4, 150 mM NaCl, 0.25% NP-40, 1 mM DTT, and CPIs). Five hundred picomoles of mycG4 ssDNA oligonucleotide was added to each reaction (no DNA was added to duplicate control reactions). Reactions were incubated for 90 min at 4 °C while rotating end-over-end. Formaldehyde was added to a 1% final concentration, and reactions were incubated for 10 min at 30 °C. Cross-linking was quenched by adding glycine to a final concentration of 12.5 mM and incubating for 5 min at 30 °C. Twenty microliters of

streptavidin–sepharose bead slurry was added to each reaction, and bead reactions were incubated for 30 min at 4 °C with rotation. Each reaction was washed three times with 8 M urea prepared in 50 mM ammonium bicarbonate. Each reaction was resuspended in 100 µL of 2 M urea with 50 mM ammonium bicarbonate and 10 mM DTT. Samples were incubated for 30 min at room temperature with shaking on a tabletop shaker. Indole-3-acetic acid was added to each reaction to 50 mM and incubated for 10 min in the dark while shaking. Trypsin of 2.5 µg was added to each sample, and reactions were incubated overnight at room temperature while shaking. Digested samples were washed three times with 8 M urea plus 50 mM ammonium bicarbonate and three times with 50 mM ammonium bicarbonate to remove digested peptides not cross-linked to DNA oligonucleotides. One hundred microliters of 50 mM ammonium bicarbonate was added to each sample, and cross-links were reversed by incubating at 70 °C for 90 min. The supernatant was collected and transferred to a new tube. Fresh trypsin of 0.25 µg was added to each sample. Samples were incubated at room temperature for 4 h with shaking, prepared on stageTips, and labeled by dimethyl chemical labeling on stageTips as described previously[40,41].

Isobaric labeling was performed using the 10-plex TMT system (Thermo)[15]. TMT reagent of 0.8 mg for each reporter mass was resuspended in 100 µL anhydrous acetonitrile. Resuspended TMT reagent of 10 µL was then added to the corresponding sample. Reactions were incubated for 1 h in the dark before quenching for 30 min with 100 mM Tris, pH 8.0. All ten pulldowns corresponding to all ten oligonucleotide titration points labeled by the corresponding TMT reagent were pooled into one Eppendorf tube, acidified with trifluoroacetic acid, and desalted for mass spectrometry analysis by the C18 StageTip method[42].

**Preparation of modified nucleosomes.** Recombinant human core histone proteins were expressed in *E. coli* BL21(DE3)/RIL cells from pET21b(+) (Novagen) vectors and purified by denaturing gel filtration and ion exchange chromatography essentially as described[43]. Truncated human H3.1Δ1-31T32C protein for native chemical ligation of modified histone H3 variants was expressed in BL21(DE3)/RIL cells and purified as described[44]. Native chemical ligations were carried out in 550 µl of degassed native chemical ligation buffer (200 mM KPO4, 2 mM EDTA, 6 M guanidine HCl) containing 1 mg of modified H3.1 amino acids 1–31 thioester peptide (Cambridge Peptides), 4 mg of truncated H3.1Δ1-31T32C, 12.5 mg 4-mercaptophenylacetic acid, and 10 mg TCEP as a reducing agent at a pH of 7.5. The reactions were incubated overnight at 40 °C and quenched by addition of 60 µl of 1 M DTT and 700 µl of 0.5% acetic acid. After a centrifugation step to remove precipitates, the ligation reactions were directly loaded and purified on a reversed phase chromatography column (Perkin Elmer Aquapore RP-300 C8 250 × 4.6 mm² i.d.) using a gradient of 45–55% B (buffer A: 0.1% trifluoroacetic acid (TFA) in water; B: 90% acetonitrile, 0.1% TFA) over 10 column volumes. Positive fractions containing ligated full-length histone H3.1 were then combined and lyophilized. Histone octamers were refolded from the purified histones and assembled into nucleosomes with biotinylated DNA via salt deposition dialysis as described[43]. Biotinylated nucleosomal DNAs containing either one (mono-nucleosomes) or two 601 nucleosome positioning sequences[45] separated by a 50 bp linker (di-nucleosomes) were prepared as described[44]. Di-nucleosomes were assembled in the presence of MMTV A competitor DNA and a slight excess of octamers as described for longer chromatin arrays to ensure saturation of the 601 repeats[46]. The reconstituted nucleosomes were then immobilized on magnetic streptavidin beads (Dynabeads MyOne Steptavidin T1) via the biotinylated DNA, washed to remove MMTV A competitor DNA and MMTV A nucleosomes (in the case of di-nucleosomes), and directly used for affinity pull-down reactions as described above. Nucleosome quality control checks are shown in Supplementary Fig. 9.

**Mass spectrometry analysis.** Samples containing labeled peptides were eluted from StageTips with buffer B (80% acetonitrile, 0.1% formic acid), concentrated to 5 µL by SpeedVac centrifugation at room temperature, and resuspended to 12 µL in buffer A (0.1% formic acid). Samples were separated by liquid chromatography using an Easy-nLC 1000 system (Thermo). For our first SP/KLF replicate, we used a modified gradient from 7 to 15% buffer B over 5 min, from 15 to 35% buffer B over 174 min, from 35 to 50% buffer B over 5 min, and finally from 50 to 95% buffer B in 1 min followed by 5 min hold at 95% buffer B. For all other replicates and experiments, the gradient was the same, with the exception of the 15–35% buffer B gradient extending over 214 min.

Mass spectrometry analysis was performed on a Thermo Fusion Tribrid instrument using the built-in Thermo SPS-MS3 method, with a modified nano-HPLC gradient as described above[16,17]. Briefly, full MS scans were collected in the orbitrap at 120,000 resolution in a scan range from 380 to 1500 *m/z*. We used an AGC target of 2.0e5, and a maximum injection time of 50 ms. Peaks were selected for MS2 based on selection criteria of charge state 2–7 and an intensity threshold of 5.0e3. Dynamic exclusion was enabled, with peaks excluded after 1 scan for a duration of 70 s in a ±10 ppm window. MS2 was conducted in top speed data-dependent acquisition mode, with precursor priority given based on highest intensity. MS2 scans were performed after isolation in the quadrupole using an isolation window of 0.7 *m/z* units. We used CID activation at a collision energy of 35% for fragmentation. MS2 detection was performed in the ion trap with an AGC target of 1.0e4 and a maximum injection time of 50 ms. MS2 precursors in the mass range 400–1200 *m/z* were selected for MS3 analysis using the Thermo TMT reagent

isobaric tag loss exclusion property and excluding MS2 precursor ions 18 $m/z$ units low and 5 $m/z$ units high. MS3 selection was conducted in top 10 data-dependent acquisition mode giving the most intense ions the highest precursor priority. MS3 ions were selected with synchronous precursor selection activated for 10 precursors. MS and MS2 isolation windows were set to 2 $m/z$. HCD activation was used at a collision energy of 65%. Fragment ions were detected in the orbitrap with 60,000 resolution in the scan range 120–500 $m/z$. For MS3, we used an AGC target of 1.0e5 and a maximum injection time of 120 ms.

Mass spectrometry analysis of formaldehyde cross-linking experiments was performed using a 2 h gradient with chromatography and instrument settings as reported previously[41].

**Computational identification and quantification of proteins**. Spectral matching to peptides, grouping of peptide identifications into proteins, and isobaric label quantification were performed using Proteome Discoverer 2.1 (Thermo). We used the built-in processing workflow "PWF_Fusion_Reporter_Ba-sed_Quan_SPS_MS3_SequestHT_Percolator" and the built-in consensus workflow "CWF_Comprehensive_Enhanced_Annotation_Quan_Results," both with default settings. We used the TMT 10-plex quantification method with the 131 mass set as the control channel. In our workflow, the 131 reporter mass always corresponded to the titration point with the highest bait concentration (3 μM), with each sequentially lighter reporter tag corresponding to a three-fold dilution of the next highest bait concentration. For the Sequest HT search, database parameters were enzymatic digestion with trypsin allowing two missed cleavages, a minimum peptide length of 6 amino acids and a maximum peptide length of 144 amino acids. Our search was performed against the uniprot curated human proteome (downloaded December 2015). We used a precursor mass tolerance of 10 ppm and a fragment mass tolerance of 0.6 Da. Cysteine carbamidomethylation was included as a static modification (57.021 Da), while methionine oxidation (15.995 Da) and protein N-terminal acetylation (42.011 Da) were included as dynamic modifications. We included the 6-plex TMT reagent mass (229.163 Da) as a dynamic modification on lysine, histidine, serine, and threonine, as well as the peptide N terminus. FDR filtering was performed via percolator with a strict target FDR of 0.01 and a relaxed FDR of 0.05[47]. Strict parsimony was applied for protein grouping, and unique plus razor peptides were used for quantification. Peptide quantification normalization was applied based on total peptide amount.

For peptide searching taking into account peptide phosphorylation, we considered serine, threonine, and tyrosine phosphorylation events (79.966 Da) as possible dynamic modifications. All other parameters were unchanged.

Peptide identification and quantification of dimethyl chemical labels for formaldehyde cross-linking experiments was performed using the MaxQuant software package[48] v1.6.0.1 searching against the UniProt curated human proteome (released June 2017). Carbamidomethylation was included as a fixed modification and methionine oxidation and protein N-terminal acetylation were included as variable modifications. Requantification was selected as a quantification parameter. Normalized peptide ratios were transformed to log 2, and replicates were plotted against each other in two dimensions. Outliers were called based on inter-quartile ranges using an inter-quartile range of 1.5 in each replicate as a cutoff value.

**HeLa nuclear lysate absolute proteome quantification**. Absolute abundances of proteins in HeLa nuclear lysate were reported previously (Supplementary Data 2)[22]. Quantification was based on the iBAQ method as described previously using MaxQuant version 1.2.2.5[48–51].

**Fitting of binding parameters and statistical analysis**. To calculate protein binding parameters, we fit a Hill-like curve of the form:

$$\theta = \frac{1}{\left(\frac{K_d^{App}}{[L]}\right)^n + 1}, \qquad (1)$$

where $\theta$ represents the fraction of protein bound, $[L]$ represents the concentration of bait, $K_d^{App}$ represents the apparent dissociation constant at which half the protein is bound to a bait molecule, and $n$ is the Hill coefficient describing the rate at which binding saturates. $\theta$ was observed by calculating the normalized ratio of each titration point to the 131 reporter ion signal (representing a 3 μM bait concentration), implicitly assuming that for each protein that shows $K_d^{App}$ values in the nanomolar range we would essentially saturate binding at this titration point. Then, the signal from the 131 reporter ion represented the complete bait binding population of the entire sample, so the signal from each titration point relative to the 131 reporter ion represented the fraction bound of the total binding population. $[L]$ was known from the experimental design. We fit the parameters $K_d^{App}$ and $n$ using non-linear least squares using the mean $\theta$ of each triplicate (SP/KLF oligo experiments and nucleosome experiments) or duplicate (motif survey experiments including mycG4 experiments). Proteins were considered for further analysis only if $\theta$ was measured for each titration point of each replicate. $\theta$ values from MS3 quantification in Proteome Discoverer were normalized by min–max scaling between 0 and 1. We used initial parameter estimates of $K_d^{App} = 100$ and $n = 1$. After fitting, our data were filtered first based on the goodness of fit of the Hill-like

curve. We required a $r^2$ value of 0.95 (0.9 in nucleosome experiments) for a linear regression between the fit binding model and the measured data, as well as a predicted fraction bound of <0.25 for the lowest titration point and >0.75 for the highest titration point. In other words, fit binding curves should match the data well to avoid spurious fitting, and the binding curve should nearly saturate on both sides based on the assumptions in the workflow described above. For SP/KLF competition experiments, binding parameters were fit using a Hill-like curve essentially as described above, using an $r^2$ cutoff of 0.95, to estimate $IC_{50}$ values. These $IC_{50}$ values were then converted to $K_d^{App}$ values using the Cheng–Prusoff correction:[8]

$$K_d^{App}\_Free = \frac{IC_{50}}{1 + \frac{200}{K_d^{App}\_Immobilized}}, \qquad (2)$$

where $K_d^{App}\_Free$ is the calculated $K_d^{App}$ for either the wild-type or the mutated SP/KLF oligonucleotide, $IC_{50}$ is the fit $IC_{50}$ value from the competition experiment, and $K_d^{App}\_Immobilized$ is the $K_d^{App}$ value calculated from the immobilized SP/KLF wild-type oligonucleotide experiment. $K_d^{App}$ values identified in this study are listed in Supplementary Data 1. Binding curves were plotted as the mean of replicates plus and minus error bars representing the standard error of the mean calculated using bootstrapping in the python package seaborn. Clustering of protein binding profiles was performed by hierarchical agglomerative clustering, and the mean $K_d^{App}$ value of each sample-cluster combination was plotted in a heat-map. The number of clusters was selected by manual inspection. For statistical comparisons between multiple members of a complex or complexes, we performed a two-sided $t$ test treating all measured $K_d^{App}$ values for that complex or paralog group as sample populations as described in the main text and figure legends.

**Comparison of $K_d^{App}$ and motif score**. To compare fitted $K_d^{App}$ values with motif scores from genome-wide binding models, we first collected all vertebrate JASPAR motifs for factors both with measured $K_d^{App}$ values and identified in control experiments with mutated SP/KLF oligonucleotide as sequence-specific. To account for intrinsic differences in length and information content of the different motifs, we calculated the normalized motif score, defined here as the maximum motif score for the oligo sequence used divided by the patser motif significance threshold in biopython[52].

**G-quadruplex enrichment in PRC2 and NuRD ChIP-seq peaks**. We compared publically available ChIP-seq data for SWI/SNF, PRC2, and NuRD subunits[29,30] with publically available G4-sequencing data[31] (Supplementary Table 3). All sequencing datasets were mapped to the human genome build hg38 using the UCSC genome browser liftOver tool[53]. All G4 peaks from the plus and minus strand with overlapping coordinates were combined using bedtools[54]. We used automated permutation-based testing with pybedtools[55] to look for significant correlation between ChIP-seq peaks and G4-seq peaks. We randomized ChIP-seq peaks 1000 times over the genome and each time measured the peak intersection with G4-sequencing peaks. We then calculated an empirical $p$ value by comparing the number of true intersected peaks between SWI/SNF, PRC2, or NuRD subunits and G4-seq peaks with these 1000 randomized intersections.

**Protein cloning, expression, and western blotting**. DNA pulldowns were performed as described above for western blotting. For the recombinant SP3 and SP3+HeLa spike-in pulldowns, 50 ng SP3 protein/pulldown was used, and the protein binding step was performed in protein binding buffer supplemented with 10 ng/μL bovine serum albumin (BSA) (final concentration), 10 μM ZnCl₂, and 10% glycerol. After the washing steps, 20 μL sample buffer (1× Laemmli buffer prepared in a final concentration of 6 M urea) was added to each sample. Samples were incubated for 30 min at 37 °C and resolved by denaturing polyacrylamide gel electrophoresis. Proteins were transferred to either nitrocellulose or PVDF membranes by semi-dry transfer using the Bio-Rad Trans-Blot Turbo transfer system. Membranes were blocked in 5% milk for 30 min, and proteins were imaged by immunoblotting using the Thermo Super Signal Pico PLUS chemiluminescent substrate.

The recombinant N terminally GST-tagged KLF4-ZF domain was expressed in BL21 Rosetta (DE3) bacterial cells using 1 mM isopropyl β-D-1-thiogalactopyranoside induction performed overnight at 16 °C. Two hundred milliliters of cell culture were collected and incubated in 10 mL lysis buffer (phosphate-buffered saline (PBS), 0.1% NP-40, 1 mM DTT, CPIs, 10 μM ZnCl₂, 2.5 μM MgCl₂, 0.25 mg/mL lysozyme, 2 μL benzonase (>500 U)) on ice for 10 min. Cells were lysed by 30 s rounds of sonication followed by 30 s incubation on ice until the lysate cleared. Lysates were centrifuged at 4600 × $g$ for 30 min at 4 °C and the soluble supernatant was immediately added to 150 μL glutathione-agarose beads that had been prewashed 1× with DBB (Pierce). Ethidium bromide was added to the supernatant to a final concentration of 20 μg/mL, and lysates were incubated with beads for 1 h while rotating end-over-end at 4 °C. Beads were washed six times with DBB (1 M NaCl) and three times with PBS. GST-tagged proteins were eluted from the beads with 50 mM reduced glutathione in PBS. Eluted proteins were dialyzed into PBS (+10 μM ZnCl₂) twice over 2 h at 4 °C before a final dialysis overnight into PBS (+10 μM ZnCl₂) at 4 °C using

Slide-A-Lyzer dialysis cassettes (10,000 molecular weight cut-off, 0.1–0.5 mL volume, Thermo). NP-40 was added to 0.1% and DDT was added to 1 mM, while NaCl was adjusted to 400 mM in PBS for storage. Protein concentration was measured by UV absorbance at 280 nM and by the Bradford assay. KLF4-ZF experiments were repeated after dialyzing GST-KLF4-ZF into PBS with no additions, and similar results were observed.

The PHF10 C-terminal double PHD finger domain and the SMARCB1 N-terminal WHD were N terminally GST-tagged during ligation-based cloning, expressed in BL21 Rosetta (DE3) bacteria, and used for H3 peptide pulldowns as described previously[56]. Antibodies and dilutions used in this study are listed in Supplementary Table 2, and raw gel images are shown in Supplementary Fig. 11.

**Electrophoretic mobility shift assays**. For EMSA experiments probing for endogenous protein from HeLa lysates, only PBB was used as the incubation buffer. DNA oligos were diluted in PBB as described above and incubated with ~0.75 µg/µL HeLa nuclear lysate for 15 min at room temperature. Binding reactions were then resolved by native polyacrylamide gel electrophoresis in 0.5× TBE (Tris-borate-EDTA) running buffer. Protein transfer and immunoblotting was performed as described above. For EMSAs using purified recombinant SP3, the oligo dilution and protein binding steps were performed in protein binding buffer supplemented with 10 ng/µL BSA (final concentration), 10 µM $ZnCl_2$, and 10% glycerol. One hundred nanograms of SP3 protein per reaction was used for the recombinant SP3 EMSA. Recombinant proteins used in this study are listed in Supplementary Table 2.

For KLF4-ZF EMSA experiments, proteins were prepared with DNA based on molar concentration as shown in Supplementary Fig. 3C in PBS. Binding reactions were incubated for 15 min at room temperature and resolved on a 1% agarose gel using 0.5× TAE (Tris-acetate-EDTA) running buffer. The agarose gel was washed for 5 min in distilled water, stained for 30 min with 0.5 µg/mL ethidium bromide, and destained for 5 min in distilled water before imaging.

**Fluorescence polarization and fluorescence intensity assays**. Fluorescence polarization and intensity experiments were performed largely as described previously[19,20]. SP/KLF wild-type oligonucleotides were ordered with a 5′ Cy5 fluorescent label on each strand (IDT). Labeled oligonucleotides were annealed as described above and diluted to 2 nM in PBS. Recombinant KLF4-ZF was diluted using PBS to a three-fold dilution series of ten concentrations from 0.3 nM to 6 µM in a Greiner black, 96-well non-binding microplate. Recombinant KLF4-ZF and labeled oligonucleotides were mixed at a ratio of 1:1 in a final volume of 200 µL, with labeled oligonucleotides at a final concentration of 1 nM and recombinant KLF4-ZF at a final concentration range of 0.15 nM to 3 µM. Binding reactions were incubated for 20 min at room temperature, and fluorescence polarization and intensity were measured at 25 °C on a Tecan Spark 10 M microplate reader. Baseline polarization was calibrated based on protein-free reference samples to 50 mP. Wells were measured with 200 flashes per well and a 1 s settling time per sample. Binding assays were performed in triplicate, and binding parameters were calculated and plotted as described above using a Hill-like function.

**Data availability**. The mass spectrometry proteomics data have been deposited to the ProteomeXchange Consortium via the PRIDE partner repository[57] with the dataset identifier PXD007132. All other relevant data are available from the corresponding authors.

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

## Acknowledgements

We thank all members and students of the Vermeulen group for fruitful scientific discussion, kind input, and critical feedback. We would like to thank Dr. C.G. Spruijt for the generous gift of KLF4-ZF expressing bacterial lysate and for the GST-KLF4-ZF expression construct. We would like to thank Dr. Falk Butter for cloning the His-PHF10 construct used for cloning GST-PHF10-DPF, and Jasmin Mecinovic for help with quantitative protein binding assays. This work is part of the Oncode Institute which is partly financed by the Dutch Cancer Society (KWF) and was funded by the gravitation program CancerGenomiCs.nl from the Netherlands Organization for Scientific Research (NWO). Funding for M.M.M. was provided by a grant from the Marie Curie Initial Training Network (ITN) DevCom (FP7, grant number 607142). Funding for T.B was provided by the Medical Research Council (Grant No. MC_UP_1102/2) and the European Research Council (ERC StG, no. 309952).

## Author contributions

M.V., T.B., and M.M.M. conceived the study and designed experiments. C.G. performed $K_d^{App}$ experiments with DNA and nucleosome baits and analyzed data. B.M.F. and N.V. N. produced and assembled recombinant (modified) nucleosomes and performed related quality control checks. M.M.M. performed $K_d^{App}$ experiments, analyzed data, and wrote the manuscript with input from all co-authors. M.V. and T.B. supervised the project and provided critical feedback at all stages.

## Additional information

**Competing interests:** The authors declare no competing interests.

