## [Peer Review File · Nature Communications]

Reviewer #1:

Remarks to the Author:

In the manuscript 'Proteome-wide affinity quantification by mass spectrometry' Vermeulen and co-workers describe a quantitative proteomics workflow that allows quantifying interactions of protein with DNA and nucleosomes. The approach is benchmarked with the Sp/KLF consensus GC box and then applied to a number of DNA motifs and mono-, di- and tri-nucleosomes.

The so called paqman workflow resembles proteomics approaches for characterizing the interactions of proteins with small molecule ligands and may not be entirely novel. The application to DNA and nucleosomes, however, makes the paqman workflow, rather distinct and the application examples nicely demonstrate its large potential for proteome-wide interaction studies. This reviewer supports publication of the manuscript under consideration once points listed below have been addressed.

- Technical validation and information on selectivity and sensitivity of the approach are not sufficiently addressed. Controls such as scrambled sequences and competition by spiking excess concentrations into nuclear extracts would be instructive at least in benchmarking experiments. It would also be useful to spend a few words on protein background and how the approach distinguishes specific binding from unspecific binding to the streptavidin sepharose.
- Validation of the determined dissociation constants: Some more reference data based on independent methods would be required to give confidence that the determined apparent dissociation constants are reasonably accurate. In their calculations, the authors require that binding equilibrium is fully established (no time-dependent changes) and that all DNA molecules captured on the streptavidin sepharose are available for binding to protein interactors. How is this controlled for?
- The title suggests a much wider scope than what is covered in the manuscript, suggest to adjust
- The abstract comes across as rather defensive. Most of the text is devoted to arguing why previous work was insufficient. The authors might consider giving the abstract a more positive spin, describing what their approach enables.
- Suggest to add data points to all graphs containing fitted curves at least as supplementary material
- Sharma et al Nature Methods 2009 essentially described, how affinity constants of immobilized peptides and other compounds can be determined by simple consecutive binding experiments with cell extracts. I wonder why the authors decided to use the 10-point bead titration approach and how this compared to the method by Sharma that the authors were surely aware of.

Reviewer #2:

Remarks to the Author:

The manuscript by Makowski et al. describes the paqMan method to quantify protein-DNA and protein-nucleosome apparent binding affinities from nuclear extract. The paqMan uses 10-serial-diluted DNA as a bait, labeled with 10 different reagents to allow a single mass spectrometry run, producing a Hill-like curve and K_{dapp} values for each bound protein. The authors demonstrated the method with the well-characterized Sp/KLF consensus GC-box, comparing side by side with gel-shift assays. Then they analyzed other DNA motifs binding affinities to nuclear proteins using paqman. They went on to study the difference in affinities between proteins and 3 different kinds of mono- and di-nucleosomes, unmodified, H3K4me3, and H3K9ac14ac. The manuscript is well-written, and presents new and valuable information which will be beneficial to researchers in the chromatin field. Therefore, I recommend this work as a publication in Nature Communication, but some revisions may improve the work better.

Major comment

1) Where is the 'kinetic binding detected' mentioned in pg 2, line 23(2nd paragraph of the main text) for 'a number of other Sp/KLF family factors'? I cannot locate any time-dependent binding assay in the manuscript. The 'binding kinetics' is mentioned again in pg 3, line 7.

2) The authors wrote inaccurate inferences of the affinity. In the main text (pg 3, line 26) explaining LiCl- and PhenDC3- inhibition of binding to G4, it was mentioned K_{dapp} of PRC2 and NuRD G4 was decreased. This should be changed to 'increased'.

Also, in the main text (pg 4, line 4) K_{dapp} (~66nM) for SWI/SNF binding to H3K9AcK14Ac-di-nucleosome is lower, and that to H3K4me3-di-nucleosomes is higher.

The same goes for K_{dapp} for BPTF to H3K9AcK14Ac-di-nucleosome, which is lower, not higher than that to H3K4me3-di-nucleosome.

3) For Sup.Fig 13C, there should be a control/standard for the western blot.

4) How did the authors check the 'positive fractions' following the purification of ligated H3.1? It would be better to add a supplementing data confirming the ligation.

5) Did the authors make sure of the mono- or di-nucleosome formation without free-DNA? Native gel confirming this needs to be presented.

Minor comment

1) Any perspective in the difference between K_d (from a gel-shift) and K_{dapp} from paqMan for SP3 but not for KLF4 binding? Is there any evidence that SP3 has competitors for binding but KLF4 doesn't? I noticed KLF4 is the farthest one in Sup.Fig 5C showing the correlation between K_{dapp} and the JASPAR motif score (The publication for JASPAR may need to be cited). Does it mean anything?

2) The cartoon in Fig.3A is somewhat misleading because there is H3K4me3K9K14 Di-Nucleosome instead of H3K4me3 Di-Nucleosome, which is what the authors studied.

3) In methods, under Fitting of binding parameters and statistical analysis, at the start of 2nd paragraph, our data -> our data?

Reviewer #3:

Remarks to the Author:

Review: "Proteome-wide affinity quantification by mass spectrometry" by Makowski and colleagues

Makowski et al. combine titration of oligonucleotide affinity purifications with multiplexed quantitative mass spectrometry using TMT labeling to infer the apparent binding affinities of proteins to DNA and nucleosomes in the context of nuclear extracts. The authors nicely show the applicability of their method for double stranded and single stranded DNA templates as well as for nucleosomes. The experiments are executed at high level and with appropriate controls. The resulting affinities appear to be quite robust across replicates and in agreement with a wealth of literature information. Whereas oligonucleotide affinity purification and TMT labeling are quite established methods the authors convincingly demonstrate that the combination of these methods in paqMan results in robust data filtering on the basis of the obtained Hill-like curves and greatly facilitates data interpretation compared to previous approaches where simply lists of proteins have been obtained. Overall, I believe that this approach could be of interest to a broader group of biologists studying the organization and regulation of DNA associated protein complexes. Whether this method will become routine also for addressing KDapp at larger scale remains to be seen since the associated costs for mass spectrometry and the commercial labeling reagents are quite significant. At the current state I think the authors should therefore tone down a bit their broad title claim "proteome-wide affinity quantification by mass spectrometry" as it may raise expectations that currently cannot be met.

The study would profit by addressing the following major and minor points:

Major points

1. We do not agree with the claim in the abstract: "few truly quantitative workflows exist". This statement is too strong and should be revised to reflect the numerous contributions on quantitative interaction proteomics in the literature.
2. The authors should more clearly explain why knowing the absolute KDapp would be beneficial for the community.
3. Their study does not distinguish between "direct" and "indirect" affinities of proteins to DNA. Proteins acting as subunits of complexes can therefore show apparent affinities to DNA indirectly through their complex formation with other partners such as DNA binding transcription factors that directly bind to DNA. There are many examples for this in the provided data. For example the authors measured the apparent affinities for proteins to bind to the TEAD ds oligonucleotide in Fig.2. In the supplementary data table they have listed the corresponding affinity for the DNA binding transcription factor TEAD4. A similar affinity was also measured for YAP1 a transcriptional coactivator that cannot bind DNA directly but forms complexes with TEAD transcription factors. This results clearly shows that their method estimates Kdapp in the context of protein complexes. However this is not sufficiently analyzed and discussed in their manuscript. I think the study would benefit if the authors would systematically integrate public protein interaction data and functional annotations with their measured affinities. This would allow for more precise hypotheses on the organization of DNA binding protein complexes on the basis of Paqman data. The same holds true

of course for the nucleosome experiments: it would be informative to distinguish proteins that bind directly to nucleosomal histones from those that bind indirectly. I would interesting to see if the clustering of the KDapp could enrich for complex subunits.

4. On page 3: the term “measurable binding curves” should be precisely defined. What are the criteria applied to include or exclude a measured curve?

5. It is known that posttranslational modifications (PTMs) represent a major mechanism to control nuclear transport as well as DNA complex formation. Is there evidence in the data that PTMs affect the shape of the Hill-like curve of a protein? Does a specific PTM affect the the KDapp (increase or decrease)? Is the Hill-like curve at all affected if PTMs are included in the analysis? The authors may therefore reanalyze their data in a peptide centric mode to define whether differentially modified (i.e. phosphorylated) peptides from the same protein display different KDapp, which could hint at a potential mechanism of regulating binding affinities.

Minor points

1. supplementary Fig. 2, no negative control i.e. with mutated SP1 binding sites are included.

2. On page 5 in the section “affinity purification and sample preparation” duplication of “while shaking”

3. In supplementary figure 13 the annotation of the graph is not consistent with the protein annotation used in the text. C14orf49 instead of C17orf49 mentioned in the main text and the supplementary figure description.

Reviewers' comments:

Reviewer #1 (Remarks to the Author):

In the manuscript 'Proteome-wide affinity quantification by mass spectrometry' Vermeulen and co-workers describe a quantitative proteomics workflow that allows quantifying interactions of protein with DNA and nucleosomes. The approach is benchmarked with the Sp/KLF consensus GC box and then applied to a number of DNA motifs and mono-, di- and tri-nucleosomes.

The so called paqman workflow resembles proteomics approaches for characterizing the interactions of proteins with small molecule ligands and may not be entirely novel. The application to DNA and nucleosomes, however, makes the paqman workflow, rather distinct and the application examples nicely demonstrate its large potential for proteome-wide interaction studies.

This reviewer supports publication of the manuscript under consideration once points listed below have been addressed.

- Technical validation and information on selectivity and sensitivity of the approach are not sufficiently addressed. Controls such as scrambled sequences and competition by spiking excess concentrations into nuclear extracts would be instructive at least in benchmarking experiments.

We have now provided these controls via MS experiments with mutated SP/KLF sequence and conventional competition experiments with both wild-type and mutated SP/KLF sequence. We have provided a western blot control that binding of SP/KLF factors to the wild-type and not the mutated oligonucleotide is highly specific (Supplementary Fig. 2B). We also show in Supplementary Figure 4, via competition experiments and converting IC50 values to K_d^{app} values, that K_d^{app} values of sequence specific proteins are reproducible even via a completely different quantitative binding assay workflow. We have added the following discussion of these experiments to the main text:

“To further characterize the specificity and sensitivity of this assay, we conducted a series of competition experiments with free (unbiotinylated) wild-type and mutated SP/KLF oligonucleotides. In agreement with western blot analysis (Supplementary Fig. 2B), known sequence-specific SP/KLF transcription factors showed no measurable binding to the mutated SP/KLF oligonucleotide (Supplementary Fig. 4A). In competition experiments, these sequence-specific factors were similarly not competed away from immobilized wild-type oligonucleotides by free mutated oligonucleotides (Supplementary Fig. 4A). Yet, importantly, K_d^{app} values for sequence-specific proteins estimated by applying the Cheng-Prusoff correction to IC50 values from competition experiments (Supplementary Fig. 2B) were highly correlated to K_d^{app} values estimated with immobilized oligonucleotides at a near 1:1 ratio (Supplementary Fig. 4C). In contrast, non-sequence specific proteins bound to immobilized wild-type or mutated SP/KLF oligos with high correlation (Supplementary Fig. 4A). Non-sequence specific proteins were also competed from immobilized wild-type oligonucleotides by either wild-type or mutated free oligonucleotide. However,

non-sequence specific proteins displayed lower K_d^{app} values on average to either free oligonucleotide compared to immobilized oligonucleotides in competition experiments (Supplementary Fig. 4C). We attribute this to free oligonucleotides having two available blunt ends acting as substrates for some non-sequence specific proteins, while immobilized oligonucleotides only have one sterically free blunt end. This is an important consideration for future studies comparing free v. immobilized oligonucleotides and sequence-specific v. non-specific DNA binding factors.”

It would also be useful to spend a few words on protein background and how the approach distinguishes specific binding from unspecific binding to the streptavidin sepharose.

We have added the following discussion to the main text:

“We filtered out background or non-specific proteins based on the quality of fit of the Hill curve, under the assumption that background proteins would show randomly distributed ratios near 1:1 for all titration points. As such, only proteins fitting a Hill-like curve with an r-squared value greater than 0.95 were kept for downstream analysis.”

We note that, generally, we identify 1000+ proteins in each of the main experiments we describe in the manuscript. So we are effectively filtering out hundreds to thousands of background proteins by the relatively stringent filtering criterion we describe.

- Validation of the determined dissociation constants: Some more reference data based on independent methods would be required to give confidence that the determined apparent dissociation constants are reasonably accurate.

We have attempted to validate the reliability of our measurements in three main ways.

First, we believe the competition experiments described above, demonstrating that K_d^{app} values are consistent particularly for sequence-specific proteins whether derived from our initial workflow or via competition experiment derived IC50 values, validates that our general methodology gives accurate results even when an entirely different experimental setup is used.

Second, we have now performed additional experiments purifying recombinant KLF4-ZF to near homogeneity and performing fluorescence polarization and fluorescence intensity (dequenching) quantitative binding assays. In preparing recombinant KLF4-ZF purifications, we noticed a substantial amount of DNA contamination that likely affected our previous KLF4-ZF affinity purification western blot experiments. Hence, we have now removed that experiment (previously Supplementary Fig. 2D) from our current resubmission. In that experiment, we saw that KLF4-ZF bound with an K_d^{app} value near, or perhaps marginally higher than, the K_d^{app} value measured in our MS experiments. However, we now provided additional evidence that our KLF4-ZF preparations have negligible protein and DNA contamination (Supplementary Fig. 3). We see by both fluorescence polarization and fluorescence intensity measurements that KLF4-ZF binds the SP/KLF consensus motif with a K_d of ~21nM, approximately four-fold lower than our MS measurement. This is now in line with what we see by gel based

assays for endogenous and recombinant SP3 (i.e., that the recombinant protein binds with a higher affinity in quantitative binding assays when purified, but is marginally inhibited from binding in the context of nuclear lysates). We described this finding in the following addition to the main text:

“Similarly, we observed a lower K_d value for the SP/KLF motif and recombinant KLF4-ZF domain using fluorescence polarization and fluorescence de-quenching assays compared to those measured for endogenous KLF4 in lysates by mass spectrometry and gel-based assays (Supplementary Fig. 3)^{19,20}. This clearly suggests competitive inhibition between proteins in the nuclear environment as has been reported for SP1-KLF4 and SP1-KLF16 among other SP/KLF factors²¹. As such, this complex interplay between DNA binding proteins indicates K_d^{app} measurements will be useful information for uncovering interactions within transcriptional networks as they exist in the in vivo nuclear environment.”

Third, we have now analyzed and plotted all our data using the mean values of experimental replicates and showing the standard error (either triplicates for SP/KLF experiments, duplicates for motif survey experiments, or triplicates which we have newly added for all nucleosome and di-nucleosome experiments). We believe this is more conventional for biochemical assays in the literature and indicates that our data stands up very well to visual inspection. However, it has the additional benefit that we can calculate coefficients of variation for the fit of the Hill curve parameters. We have plotted this data in Fig. 1C and Fig. 3C. In our estimation, the low coefficients of variation we observe indicate that the K_d^{app} values we report are indeed reliable.

In their calculations, the authors require that binding equilibrium is fully established (no time-dependent changes) and that all DNA molecules captured on the streptavidin sepharose are available for binding to protein interactors. How is this controlled for?

In theory, 2 hours should easily be sufficient time for binding experiments to reach equilibrium particularly for high affinity interactors. Similarly, the concentrations of nuclear lysate we use are as low as is feasible for a single affinity purification and should leave an excess of DNA for nuclear proteins to bind.

However, we also tested these two assumptions very early on while setting up this workflow. We provide this data as a supplementary figure for the reviewer attached at the end of this Response to Reviewers letter. While we do not believe this data adds substantially to the main text, we would be happy to include these experiments in the manuscript at the reviewers discretion.

In “Reviewer Figure 1”, we show that after a 2 hour v. a 4 hour incubation, canonical SP/KLF binding factors show little change in signal as estimated by their ratio using dimethyl chemical labeling. Similarly, with the amount of beads we use and 500 pmol DNA, we see that protein signal remains linear with input amount even for protein amounts up to 5X the amount we use as input. This indicates there is still available DNA for protein even at 5X the amount of protein used in assays in our manuscript.

- The title suggests a much wider scope than what is covered in the manuscript, suggest to adjust
- The abstract comes across as rather defensive. Most of the text is devoted to arguing

why previous work was insufficient. The authors might consider giving the abstract a more positive spin, describing what their approach enables.

We appreciate these two criticisms from the reviewer, which were also echoed by Reviewer 3. We have revised our title to better reflect the protein-DNA and mass spectrometry aspects of our assay, but would be very open to suggestions from the reviewers and from the editor. Similarly, we have attempted to rewrite sections of our abstract and discussion to give a more positive assessment of some mass spectrometry based quantitative binding assays in the literature including the novel thermal proteome profiling method.

- Suggest to add data points to all graphs containing fitted curves at least as supplementary material

As described above, we have now revised all figures using the convention of plotting and fitting based on the mean of replicates, with error bars showing the standard error of the mean.

- Sharma et al Nature Methods 2009 essentially described, how affinity constants of immobilized peptides and other compounds can be determined by simple consecutive binding experiments with cell extracts. I wonder why the authors decided to use the 10-point bead titration approach and how this compared to the method by Sharma that the authors were surely aware of.

Indeed, we performed experiments testing the compatibility of the Daub method with DNA oligonucleotide baits early on. Although their method is both elegant and simple, we noticed that because it was a single measurement assay, it was somewhat prone to noise. However, more problematic, we noticed that in general for protein-DNA interactions including sequence-specific interactions, which are known to be high affinity (nM range), measured K_d^{app} values varied in a manner that was linearly correlated with the amount of DNA used as bait. (see Reviewer Figure 2, attached at the end of this Response to Reviewers letter), which is of course an unreasonable result. We attribute this to the fact that the Daub workflow, as the reviewer notes, uses the protein ratio between consecutive experiments to back-calculate the K_d^{app} . However, for very high affinity protein-DNA interactions (i.e. >5 times lower K_d^{app} than the amount of DNA bait used), essentially all of the free protein will bind in the first of two consecutive pulldowns. Very little protein will be remaining to bind in the second pulldown. Thus, the ratio of the two pulldowns will be very high (>10 fold). It is commonly seen in mass spectrometry experiments that very high ratios in chemical labeling experiments are often noisy and there is commonly “ratio suppression”, where very high ratios are “rounded-down” based on the dynamic quantification range of the mass spectrometer. The Daub method requires that protein ratios are highly accurate to 2-3 decimal places for high ratios in a single pulldown. If this criterion cannot be met, we believe the result will be the linear correlation between DNA input concentration and measured K_d^{app} that we observe. Therefore, although the Daub paper is an elegant and important

contribution in the literature, we believe our assay produces more reliable quantification at least for high affinity protein-DNA interactions.

Reviewer #2 (Remarks to the Author):

The manuscript by Makowski et al. describes the paqMan method to quantify protein-DNA and protein-nucleosome apparent binding affinities from nuclear extract. The paqMan uses 10-serial-diluted DNA as a bait, labeled with 10 different reagents to allow a single mass spectrometry run, producing a Hill-like curve and K_{dapp} values for each bound protein. The authors demonstrated the method with the well-characterized Sp/KLF consensus GC-box, comparing side by side with gel-shift assays. Then they analyzed other DNA motifs binding affinities to nuclear proteins using paqman. They went on to study the difference in affinities between proteins and 3 different kinds of mono- and di-nucleosomes, unmodified, H3K4me3, and H3K9ac14ac. The manuscript is well-written, and presents new and valuable information which will be beneficial to researchers in the chromatin field. Therefore, I recommend this work as a publication in Nature Communication, but some revisions may improve the work better.

Major comment

1) Where is the 'kinetic binding detected' mentioned in pg 2, line 23(2nd paragraph of the main text) for 'a number of other Sp/KLF family factors'? I cannot locate any time-dependent binding assay in the manuscript. The 'binding kinetics' is mentioned again in pg 3, line 7.

We have corrected any references to “kinetic” binding. Our assay, of course, assesses equilibrium state binding and does not include kinetic, time-dependent parameters.

2) The authors wrote inaccurate inferences of the affinity. In the main text (pg 3, line 26) explaining LiCl- and PhendC3- inhibition of binding to G4, it was mentioned K_{dapp} of PRC2 and NuRD G4 was decreased. This should be changed to 'increased'. Also, in the main text (pg 4, line 4) K_{dapp} (~66nM) for SWI/SNF binding to H3K9AcK14Ac-di-nucleosome is lower, and that to H3K4me3-di-nucleosomes is higher.

The same goes for K_{dapp} for BPTF to H3K9AcK14Ac-di-nucleosome, which is lower, not higher than that to H3K4me3-di-nucleosome.

We have corrected these errors in the text, specifying that lower K_d^{app} equals higher affinity and vice versa.

3) For Sup.Fig 13C, there should be a control/standard for the western blot.

For the figure in question (now Supplementary Figure 10C) we now use the SMARCB1-WHD domain as an additional control showing that the acetylation specific histone H3 binding of PHF10-DPF is not observed in a different SWI/SNF subunit. We have also included PHF10-DPF as a control for SMARCB1-WHD mycG4 binding in Supplementary Fig. 8F.

- 4) How did the authors check the 'positive fractions' following the purification of ligated H3.1? It would be better to add a supplementing data confirming the ligation.
- 5) Did the authors make sure of the mono- or di-nucleosome formation without free-DNA? Native gel confirming this needs to be presented.

We have now included the requested nucleosome ligation and assembly quality control checks as Supplementary Fig. 9.

Minor comment

- 1) Any perspective in the difference between K_d (from a gel-shift) and K_{dapp} from paqMan for SP3 but not for KLF4 binding? Is there any evidence that SP3 has competitors for binding but KLF4 doesn't? I noticed KLF4 is the farthest one in Sup.Fig 5C showing the correlation between K_{dapp} and the JASPAR motif score (The publication for JASPAR may need to be cited). Does it mean anything?

As described above in our response to Reviewer 1, point 2, we now provide a detailed characterization of the recombinant KLF4-ZF with the SP/KLF oligonucleotide including fluorescence polarization and intensity assays. To summarize the above argument briefly, we now see that KLF4-ZF binds with an approximately four-fold lower K_d^{app} in recombinant fluorescence assays than we see via MS, which is approximately in line with what we see with SP3. Likely a similar competition effect that might be affecting SP3 in lysates could also affect KLF4. We have added this data as Supplementary Fig. 3 and have discussed this finding briefly in the main text.

- 2) The cartoon in Fig.3A is somewhat misleading because there is H3K4me3K9K14 Di-Nucleosome instead of H3K4me3 Di-Nucleosome, which is what the authors studied.

We have now corrected this in Fig. 3.

- 3) In methods, under Fitting of binding parameters and statistical analysis, at the start of 2nd paragraph, our date -> our data?

We have corrected this error.

Reviewer #3 (Remarks to the Author):

Review: "Proteome-wide affinity quantification by mass spectrometry" by Makowski and colleagues

Makowski et al. combine titration of oligonucleotide affinity purifications with multiplexed quantitative mass spectrometry using TMT labeling to infer the apparent binding affinities of proteins to DNA and nucleosomes in the context of nuclear extracts. The authors nicely show the applicability of their method for double stranded and single stranded DNA templates as well as for nucleosomes. The experiments are executed at high level and with appropriate controls. The resulting affinities appear to be quite robust across replicates and in agreement with a wealth of literature information. Whereas oligonucleotide affinity purification and TMT labeling are quite established methods the authors convincingly demonstrate that the combination of these methods in paqMan results in robust data filtering on the basis of the obtained Hill-like curves and greatly facilitates data interpretation compared to previous approaches where simply lists of proteins have been obtained. Overall, I believe that this approach could be of interest to a broader group of biologists studying the organization and regulation of DNA associated protein complexes. Whether this method will become routine also for addressing K_{Dapp} at larger scale remains to be seen since the associated costs for mass spectrometry and the commercial labeling reagents are quite significant. At the current state I think the authors should therefore tone down a bit their broad title claim "proteome-wide affinity quantification by mass spectrometry" as it may raise expectations that currently cannot be met.

We have now changed the title to reflect the focus of our assay on protein-DNA interactions using mass spectrometry. As mentioned in our response to Reviewer 1, we are open to any suggestions from the reviewers or editor for a more suitable title, if necessary.

The study would profit by addressing the following major and minor points:

Major points

1. We do not agree with the claim in the abstract: "few truly quantitative workflows exist". This statement is too strong and should be revised to reflect the numerous contributions on quantitative interaction proteomics in the literature.

We have rewritten sections of our abstract and discussion to emphasize more positively the important contributions in the literature demonstrating the feasibility of using semi-quantitative workflows to obtain absolute quantitation, including relevant citations of the thermal proteome profiling method.

2. The authors should more clearly explain why knowing the absolute K_{Dapp} would be beneficial for the community.

We have added additional discussion of this point in relevant sections, most importantly in our estimation in our discussion of SWI/SNF and ISWI subunit binding to various di-nucleosomes:

“The binding patterns we observe between different di-nucleosome baits are complex and highlight an important point and a unique benefit of measuring apparent affinities from complex lysates: we measure the average binding profile over what is likely a pool of heterogeneous multimeric protein complexes.”

“More generally, deconvolving the individual contributions of different subcomplexes with recombinant systems is indisputably critical (for example, as with SMARCB1 and its G-quadruplex interaction and PHF10-H3K9AcK14Ac binding in SWI/SNF). In the future, a data-driven strategy for deconvoluting the contributions of shared subunits within independent subcomplexes might involve a combinatorial approach utilizing many DNA or modified nucleosome baits as was demonstrated, for example, with the stoichiometry of different SET1/MLL subcomplexes²². Additionally, we argue that the ability to measure binding affinities for multimeric complexes in the complex environment of the nucleus offers an important holistic, system-wide view.”

Additionally, we have substantially revised Fig. 2B-G to include numerous examples of homo- and heter-multimeric DNA binding proteins binding to their cognate motif. We think the ability to study these multimeric interactions, as noted in this revised figure and the discussion added above, is a main reason for performing absolute quantification in a lysate context.

Finally, we suggest in the text concerning SP/KLF benchmarking experiments that:

“This clearly suggests competitive inhibition between proteins in the nuclear environment as has been reported for SP1-KLF4 and SP1-KLF16 among other SP/KLF factors²¹. As such, this complex interplay between DNA binding proteins indicates K_d^{app} measurements will be useful information for uncovering interactions within transcriptional networks as they exist in the in vivo nuclear environment.”

We believe it is relevant to the community studying protein-DNA interactions that recombinant studies might overestimate, to varying degrees, the apparent affinity of a protein-DNA interaction as it occurs in a complex lysate context.

3. Their study does not distinguish between "direct" and "indirect" affinities of proteins to DNA. Proteins acting as subunits of complexes can therefore show apparent affinities to DNA indirectly through their complex formation with other partners such as DNA binding transcription factors that directly bind to DNA. There are many examples for this in the provided data. For example the authors measured the apparent affinities for proteins to bind to the TEAD ds oligonucleotide in Fig.2. In the supplementary data table they have listed the corresponding affinity for the DNA binding transcription factor TEAD4. A similar affinity was also measured for YAP1 a transcriptional coactivator that cannot bind DNA directly but forms complexes with TEAD transcription factors. This results clearly shows that their method estimates K_{dapp} in the context of protein complexes. However this is not sufficiently analyzed and discussed in their manuscript. I think the study would benefit if the authors would systematically integrate public protein

interaction data and functional annotations with their measured affinities. This would allow for more precise hypotheses on the organization of DNA binding protein complexes on the basis of Paqman data. The same holds true of course for the nucleosome experiments: it would be informative to distinguish proteins that bind directly to nucleosomal histones from those that bind indirectly. I would be interested to see if the clustering of the KDapp could enrich for complex subunits.

We agree completely with the reviewer that our assay, “estimates Kdapp in the context of protein complexes”. As stated above, we believe this is relevant information that can be integrated with targeted recombinant studies of individual subunits or domains. The clustering approach the reviewer suggests is indeed interesting. We have taken this approach in revising Figs. 2A and 3B. As shown in Fig. 2 and mentioned in various places in Figures 2 and 3 and associated Supplementary Figures (generally in relation to example protein binding curves), we often see that shared subunits from multimeric complexes do indeed cluster together, although this is not universally true as seen from homo- and hetero-dimeric binding proteins JUNB and JUND.

On the other hand, these clusters contain relatively many members, and we often see proteins that do not participate in a multimeric complex clustering together. For example, subunits of SWI/SNF, ISWI, PRC2, and NuRD all cluster together in Fig. 2A, though these subunits almost exclusively are not shared. We believe this approach is therefore interesting in terms of grouping together proteins with shared binding patterns over a number of baits, but would be skeptical of concluding grouped proteins may represent potential “interactors”.

Finally, the remark that, “it would be informative to distinguish proteins that bind directly to nucleosomal histones from those that bind indirectly” is indeed interesting. We addressed this approach first experimentally using a formaldehyde cross-linking strategy to identify direct G4 binding proteins from chromatin remodeling or modifying complexes. This data is now shown in Supplementary Fig. 8, where we identify SMARCB1 as a SWI/SNF subunit that directly recognizes the mycG4 structure. Similarly, based on domain similarity to published double PHD finger (DPF) acetylation reader domains, we identify PHF10-DPF as an H3K9AcK14Ac reader domain. Thus, we propose an experimental strategy for identifying these “direct” interactors, as well as using the time-tested strategy of domain homology.

4. On page 3: the term "measurable binding curves" should be precisely defined. What are the criteria applied to include or exclude a measured curve?

We have modified the text as follows to try and better describe this filtering:

“Kdapp values were determined independently for each protein by fitting the parameters of a Hill-like curve using the known DNA concentrations and the observed fraction bound (Fig. 1A). We filtered out background or non-specific proteins based on the quality of fit of the Hill curve, under the assumption that background proteins would show randomly distributed ratios near 1:1 for all titration points. As such, only proteins fitting a Hill-like curve with an r-squared value greater than 0.95 were kept for downstream analysis.”

And in the methods, section “Fitting of binding parameters and statistical analysis”:

“After fitting, our data was filtered first based on the goodness of fit of the Hill-like curve. We required a r-squared value of 0.95 (0.9 in nucleosome experiments) for a linear regression between the fit binding model and the measured data, as well as a predicted fraction bound of < 0.25 for the lowest titration point and > 0.75 for the highest titration point. In other words, fit binding curves should match the data well to avoid spurious fitting, and the binding curve should nearly saturate on both sides based on the assumptions in the workflow described above.”

We note that, as described above, we now follow the convention of fitting based on the mean of replicates instead of plotting all replicates individually. Under this procedure, we were able to filter with a relatively stringent criterion of r-squared > 0.95 (0.90 for nucleosome data).

5. It is known that posttranslational modifications (PTMs) represent a major mechanism to control nuclear transport as well as DNA complex formation. Is there evidence in the data that PTMs affect the shape of the Hill-like curve of a protein? Does a specific PTM affect the the KDapp (increase or decrease)? Is the Hill-like curve at all affected if PTMs are included in the analysis? The authors may therefore reanalyze their data in a peptide centric mode to define whether differentially modified (i.e. phosphorylated) peptides from the same protein display different KDapp, which could hint at a potential mechanism of regulating binding affinities.

We have performed this analysis, and include the results as Supplementary Fig. 4D. In general, we see almost no measurable difference at least on the protein level if STY phospho-modifications are included as a variable modification in the search. In general, we identified only a relatively small number of phospho-modified peptides. It is likely specific enrichment methods or at least phosphatase inhibitors would be necessary to identify these phospho-specific differences, if present.

Minor points

1. supplementary Fig. 2, no negative control i.e. with mutated SP1 binding sites are included.

We have now included a western blot of SP1, SP3, and KLF4 with both wild-type and mutated SP/KLF oligonucleotides as Supplementary Fig. 2B. We see via western blot, and via our MS-based assay, that there is no measurable background binding for any of these factors to a mutated oligonucleotide sequence.

2. On page 5 in the section "affinity purification and sample preparation" duplication of "while shaking"

We have now corrected this error.

3. In supplementary figure 13 the annotation of the graph is not consistent with the protein annotation used in the text. C14orf49 instead of C17orf49 mentioned in the main text and the supplementary figure description.

We have now corrected this error in Fig 3D and Supplementary Fig 10D.

saturation of binding

incubation time 2h vs. 4h

Reviewer Figure 1 Protein binding is saturated and oligo binding sites are not depleted using the described assay conditions

A) DNA pull-downs were conducted as described in Hubner et al, Journal of Proteome Research, 2015, using 100pmol of annealed SP/KLF consensus oligo nucleotide incubated for 2 hours with the indicated concentration of K562 nuclear lysates. Relative abundance was estimated using triplex dimethyl chemical labeling. Abundance is calculated relative to the 100ug pull-down.

B) DNA pull-downs were conducted as described above, using 100pmol annealed SP/KLF oligonucleotide and 250ug K562 nuclear lysate. Binding reactions were incubated at 4C for either 4 hours or 2 hours, and binding ratios were calculated by dimethyl chemical labeling. Some known specific SP/KLF binding factors are indicated in red. Proteins are ranked along the x-axis from low to high based on their binding ratio.

Reviewer Figure 2 Protein-DNA KdApps for sequence specific binding factors linearly correlate with input DNA concentration
 DNA pulldowns were conducted as described in Hubner et al, Journal of Proteome Research, 2015, using the indicated amount of annealed SP/KLF consensus oligo nucleotide incubated for 2 hours with 250ug of K562 nuclear lysates. Sequential pulldowns and KdApp estimation were performed as described in Sharma et al. Nature Methods, 2009, for immobilized baits using triplex dimethyl chemical labeling. The boxplots represent calculated KdApp values for selected sequence-specific and non-specific proteins based on the Sharma et al. method of sequential incubations.

Reviewer #1:

Remarks to the Author:

In the amended manuscript, the authors have adequately addressed my initial criticism.

My congratulations to the authors for a very nice study, I fully support publication.

Just one small thing: In the amended abstract, the authors write:

'Although mass spectrometry quantitation methods are intrinsically semi-quantitative (i.e, rely on relative quantification), assay design can be leveraged to produce absolutely quantitative results.'

This sentence is quite controversial as it is not generally agreed that relative quantification is equivalent to semi-quantification. Further, there are mass spectrometric techniques that enable absolute quantification such as ICP-MS. And absolute quantification usually refers to amounts, not apparent dissociation constants. The sentence leads to confusion and does not add much to the content of the abstract. My recommendation is to leave out this rather controversial sentence and use the space to describe assay and results.

Reviewer #3:

Remarks to the Author:

The manuscript by Makowski et al. has undergone major revisions which addressed key points and open questions we and the other reviewers had raised. I congratulate all the authors to their important analytical advance towards profiling chromatin linked protein binding affinities. I have no doubts that their paper will be of broad interest once published in Nat. Comm.

REVIEWERS' COMMENTS:

We thank both reviewers once again for their constructive comments in the initial round of revision. We are happy to hear they are generally satisfied with our revisions, as we also agree that the recommended changes substantially improved the quality of our manuscript. We have responded to the remaining reviewer comments below.

Reviewer #1 (Remarks to the Author):

In the amended manuscript, the authors have adequately addressed my initial criticism.

My congratulations to the authors for a very nice study, I fully support publication.

Just one small thing: In the amended abstract, the authors write:

“Although mass spectrometry quantitation methods are intrinsically semi-quantitative (i.e, rely on relative quantification), assay design can be leveraged to produce absolutely quantitative results.”

This sentence is quite controversial as it is not generally agreed that relative quantification is equivalent to semi-quantification. Further, there are mass spectrometric techniques that enable absolute quantification such as ICP-MS. And absolute quantification usually refers to amounts, not apparent dissociation constants. The sentence leads to confusion and does not add much to the content of the abstract. My recommendation is to leave out this rather controversial sentence and use the space to describe assay and results.

We thank the reviewer for their kinds words and additional critical comments.

It was not our intention to muddy the issue of semi- v relative quantification. As we agree with the reviewer that this sentence, the general point of which we explain more clearly and with more detail in the introduction, does not add much with such a concise treatment in the abstract, we have removed it. The abstract now reads:

Interaction proteomics studies have provided fundamental insights into multimeric biomolecular assemblies and cell-scale molecular networks. Significant recent developments in mass spectrometry-based interaction proteomics have been fueled by rapid advances in label-free, isotopic, and isobaric quantitation workflows. Here, we report a quantitative protein-DNA and protein-nucleosome binding assay that uses affinity purifications from nuclear extracts coupled with isobaric chemical labeling and mass spectrometry to quantify apparent binding affinities. We use this assay with a variety of DNA and nucleosome baits to quantify apparent binding affinities of monomeric and multimeric transcription factors and chromatin remodeling complexes.

Reviewer #3 (Remarks to the Author):

The manuscript by Makowski et al. has undergone major revisions which addressed key points and open questions we and the other reviewers had raised. I congratulate all the authors to their important analytical advance towards profiling chromatin linked protein binding affinities. I have no doubts that their paper will be of broad interest once published in Nat. Comm.

We thank the reviewer for the kind words and for their help in improving the quality and content of our manuscript.